# Evidence-Based Perioperative Prevention of Postoperative Nausea and Vomiting (PONV) in Patients Undergoing Laparoscopic Bariatric Surgery: A Scoping Review

**DOI:** 10.3390/jcm14196901

**Published:** 2025-09-29

**Authors:** Piotr Mieszczański, Marcin Jurczak, Radosław Cylke, Paweł Ziemiański, Janusz Trzebicki

**Affiliations:** 11st Department of Anesthesiology and Intensive Care, Medical University of Warsaw, 02-005 Warsaw, Poland; 2Department of General Surgery and Transplantology, Medical University of Warsaw, 02-005 Warsaw, Poland

**Keywords:** PONV, nausea, vomiting, bariatric surgery, metabolic surgery, regional anesthesia, opioid-free anesthesia, multimodal analgesia, obesity

## Abstract

**Background and Objective:** Postoperative nausea and vomiting (PONV) ranks among the most common postoperative complications, affecting up to 80% of patients undergoing laparoscopic bariatric surgery. This condition negatively impacts patient comfort and well-being while also potentially delaying ambulation and increasing the risk of anastomotic and wound dehiscence. Although various interventions can mitigate the risk of PONV, none are entirely effective; therefore, combined prophylactic strategies are the standard approach. In recent years, numerous techniques and interventions have emerged; consequently, this scoping review aims to assess the current evidence regarding PONV prevention in patients undergoing laparoscopic bariatric procedures. **Methods:** This review was conducted in accordance with PRISMA guidelines and registered with OSF. A search was performed across the MEDLINE (PubMed), Scopus, Embase, and Web of Science databases. Inclusion criteria encompassed randomized controlled trials (RCTs) published up to May 2025, focusing on adult patients undergoing laparoscopic bariatric surgeries with PONV as a primary or secondary outcome. **Results:** A total of 81 studies were included in this review, encompassing a broad range of perioperative techniques, including opioid-sparing adjuvants, regional anesthesia, and pharmacological interventions. **Conclusions:** While there is general consensus and guidance advocating for a multimodal approach to PONV prevention, debates persist regarding the optimal techniques and antiemetic drug regimens to implement. Emerging evidence, particularly concerning regional anesthesia strategies and combined pharmacological prophylaxis, including novel agents, highlights the potential advantages of innovative approaches. **Highlights:** Effective management of postoperative nausea and vomiting in patients undergoing laparoscopic bariatric surgery is essential, given its impact on patient comfort, recovery, and the potential to prevent wound or anastomotic dehiscence. Although multimodal antiemetic strategies are regarded as standard, disagreements remain regarding specific measures to be adopted. New techniques and strategies, including advanced regional anesthesia techniques, pharmacological, and non-pharmacological methods, offer promising avenues for improved prophylaxis.

## 1. Introduction

Postoperative nausea and vomiting (PONV) represents a common complication following surgical procedures, affecting approximately 60–80% of patients undergoing laparoscopic bariatric surgery (LBS) [1,2]. This incidence is significantly higher compared to that observed in the general surgical patient population [3]. Numerous factors may contribute to this increased risk, including reduced gastric volume following laparoscopic sleeve gastrectomy (LSG), impaired splanchnic perfusion during pneumoperitoneum, or endocrinological factors [1,3,4,5]; however, the precise etiology of this phenomenon remains unclear.

From the patient’s point of view, PONV, along with its associated complications and effects, holds paramount clinical significance, influencing both patient comfort and safety. This spectrum of adverse factors encompasses dehydration, electrolyte disturbances, tension on sutures with the consequent risk of herniation and other wound-related complications, venous hypertension, bleeding, aspiration, and prolonged hospitalization with elevated healthcare costs [6]. Typically, PONV persists for up to 24 to 48 h following surgery [7]; however, in some cases, it may persist longer or even indicate surgical complications [8]. When nausea and vomiting occur after hospital discharge, they substantially contribute to unplanned, early readmissions, predominantly due to dehydration and electrolyte imbalances [9], thereby increasing healthcare utilization and additional costs [10]. In addition to being a potential hazard, there is compelling evidence that PONV constitutes a leading cause of patient suffering after bariatric surgery [11].

Considering the aforementioned factors, PONV prevention and treatment are one of the main goals of the guidelines for perioperative care in bariatric surgery issued by the Enhanced Recovery After Surgery (ERAS) Society. These guidelines aim to enhance patient comfort and satisfaction, which correlates with higher scores on relevant scales [12,13,14,15]. To mitigate PONV, the guidelines advocate a multimodal approach as a standard practice, including the use of dexamethasone as a supportive pharmacological intervention, shortening of perioperative fasting time, abstaining from routine nasogastric decompression, and implementing opioid-sparing strategies.

Taking into account that a vast majority of patients undergoing LBS are non-smoking women aged < 50 [16] undergoing gastrointestinal surgery [15], effective PONV prophylaxis is challenging. Moreover, although many interventions may reduce PONV incidence, none of them are entirely surefire; therefore, combined multimodal interventions are recommended [15]. The modifiable factors that can be applied to PONV prophylaxis can be divided into anesthetic techniques (including regional anesthesia), as well as both pharmacological and non-pharmacological prophylaxis and rescue therapy. Recently, numerous emerging interventions have been assessed in randomized controlled trials (RCTs). Our scoping review aims to evaluate and map the current evidence on PONV prophylaxis in patients undergoing LBS.

## 2. Materials and Methods

The scoping review was conducted in accordance with the Preferred Reporting Items for Systematic Reviews and Meta-Analyses extension for Scoping Reviews (PRISMA-ScR) [17] (Appendix A) and was registered with the Open Science Framework (OSF).

Two authors independently searched the MEDLINE (PubMed), Scopus, Embase, and Web of Science databases utilizing the keywords “bariatric surgery,” “metabolic surgery,” “laparoscopic sleeve gastrectomy,” “laparoscopic gastric bypass,” and “postoperative nausea” or “postoperative vomiting.” Furthermore, a manual review of the reference lists of the included publications was conducted to identify additional studies for possible inclusion.

The researchers employed the following sequence of research procedures: Initially, titles were screened; subsequently, abstracts and full papers were reviewed. A paper was deemed potentially relevant, and its full text was subsequently examined to determine whether its relevance could not be excluded based on its title and abstract. The full texts of all papers were subsequently assessed. In instances of disagreement, the consensus among the research supervisors was considered final.

With regard to the eligibility criteria, our review includes RCTs on PONV in laparoscopic bariatric surgery published in English. We focused on high-quality RCTs, which are essential for assessing the efficacy of interventions and providing the most dependable medical evidence, vital for practitioners. Given that the majority of RCTs are published in English, we elected to include only literature in this language due to practical and resource limitations. Studies assessing the impact of surgical techniques, their modifications, and alternative or non-conventional medical interventions on PONV were excluded from our review.

## 3. Results

The initial data search yielded 1293 articles from PubMed, 741 from Web of Science, 2341 from SCOPUS, and 2442 from Embase. A total of 6817 articles were initially retrieved. After the removal of duplicates utilizing Rayyan (Johnson & Phillips, 2018, Newport, UK), 2426 articles remained for screening.

Through a title and abstract review, 199 studies were further examined, of which 81 relevant articles, after a full-text assessment, were retrieved and included in this scoping review. The selection process is depicted in Figure 1. Data extraction was performed by two independent researchers; any disagreements were resolved by the research supervisors.

Using the Risk of Bias (ROB2) tool and the Cochrane Intervention Systematic Evaluation Manual (Cochrane Handbook for Systematic Reviews of Interventions), we conducted a methodological quality assessment of the articles that met the inclusion criteria [18]. The qualitative methodological evaluation is illustrated in Figure 2.

The relevant 81 RCTs were grouped into four groups: (1) anesthetic factors (n = 15), (2) opioid-sparing agents (n = 33), (3) pharmacological prophylaxis (n = 16), and (4) regional anesthesia techniques (n = 19). Two studies assessed both anesthetic factors and opioid-sparing agents. The relevant studies are depicted in Table 1, Table 2, Table 3 and Table 4.

We wish to emphasize that certain RCTs were designed to assess multiple interventions simultaneously. Furthermore, in the majority of these trials, antiemetic strategies were combined, potentially leading to confounding risks attributable to cointerventions. To mitigate this concern, the tables delineate both the interventions evaluated within the study groups and the antiemetics employed across both groups.

### 3.1. Anesthetic Factors in PONV Prophylaxis

15 RCTs evaluate anesthetic factors influencing PONV incidence and severity, encompassing a broad spectrum of interventions, including the selection and dosing of anesthetics, muscle relaxants, and fluid therapy. (Table 1).

**Table 1 jcm-14-06901-t001:** Anesthetic factors in PONV prophylaxis in bariatric surgery.

Author	Year	Number of Participants	Type of Surgery	Intervention Assessed	Antiemetics Used	PONV Outcome Measure	Main Results Regarding PONV
Shrivastava [19]	2025	168	LSG	Propofol TIVA, GDFT vs. control	Dexamethasone, azasetron, metoclopramide	4-point NRS within 24 h postoperatively	Propofol TIVA with GDFT decreased the incidence of PONV at 3–24 h postoperatively by 27.51%.
Elbakry [20]	2018	100	LSG	Propofol TIVA with dexmedetomidine vs. desflurane	Ondansetron	PONV score within 24 h postoperatively	Lower PONV incidence in the study group, 10% vs. 30% compared to desflurane
Ziemann-Gimmel [21]	2014	119	LYGB, LSG, LGB, revisional surgeries	Propofol TIVA, OFA vs. volatile anesthesia with opioids	Dexamethasone, droperidol, or promethazine	4-point Verbal Rating Scale in the morning after surgery	PONV Risk reduction of 46.4% in the TIVA group
Spaniolas [22]	2020	83	LSG	Propofol TIVA, aprepitant, and transdermal scopolamine vs. volatile anesthesia	Dexamethasone, ondansetron, metoclopramide, compazine	10-point Verbal Rating Scale at 1, 4, 12, 24 h postoperatively	PONV scores lower at all the time points in the study group
Aftab [23]	2019	183	LSG, LYGB	Propofol TIVA vs. desflurane	Ondansetron, metoclopramide	Visual Analog Scale at PACU, surgical ward, and 24–48 h postoperatively	No difference in PONV incidence compared with desflurane
Honca [24]	2017	61	LSG	Propofol TIVA vs. desflurane	Unspecified	PONV incidence at admission to PACU and 5, 10, 20 min postoperatively	No difference in PONV incidence compared with desflurane
De Baerdemaeker [25]	2006	50	LGB	Desflurane vs. sevoflurane	Ondansetron	Number of PONV episodes at admission to PACU and 30, 60, 120 min postoperatively	No difference in PONV incidence between desflurane and sevoflurane
Zhang [26]	2024	90	LSG	Light (BIS 50) vs. deep (BIS 35) anesthesia	Dexamethasone, palonosetron	4-point PONV grade 24 h postoperatively	No difference in PONV incidence
Castro Jr [27]	2014	88	LBS	Sugammadex vs. neostigmine	Dexamethasone, ondansetron	PONV incidence before discharge from PACU	Less PONV in the sugammadex group
Yang [28]	2024	80	LSG	Deep (PTC 1–3) neuromuscular block vs. moderate	Dexamethasone, ondansetron	PONV incidence 40 min, 24 h, 48 h postoperatively	No difference between the groups
He Huang [29]	2022	150	LSG	Deep (PTC 1–2) neuromuscular block vs. moderate	Unspecified	PONV incidence at an unspecified period	No difference between the groups
Suh [30]	2021	134	LSG, LYGB	Preoperative carbohydrate loading vs. control	Unspecified	PONV duration	Shorter duration of nausea in the study group
Cho [31]	2021	75	LSG	GDFT, SVV-guided vs. control	Prochlorperazine	11-point NRS at 0 min, 30 min, 1 h, 24 h, 48 h postoperatively	No difference between the groups
Zheng [32]	2022	137	LSG	GDFT, SVV-guided vs. control	Dexamethasone, tropisetron	4-point PONV scale at 24 h and 48 h postoperatively	GDFT effective in decreasing the incidence of PONV, 47.1% vs. 71.6% in the control group
Rossetti [8]	2014	145	LSG	Nasogastric decompression	Unspecified	PONV incidence at an unspecified period	No difference between the groups

#### 3.1.1. Volatile Anesthesia vs. Total Intravenous Anesthesia (TIVA)

In the population of patients with obesity undergoing LBS, the utilization of volatile anesthetics correlates with increased PONV incidence, whereas TIVA has a prophylactic effect [33]. Four studies regarding bariatric surgery support this conclusion [19,20,21,22]. In the RCT by Shirvastava, propofol TIVA, when compared to sevoflurane anesthesia in patients undergoing LSG, decreased the incidence of PONV by 27.51% from 3 to 24 h postoperatively, with no impact observed immediately after surgery [19]. Consistent with these findings, Elbakry et al. reported that patients in the TIVA group experienced PONV approximately 20% less frequently than those receiving desflurane anesthesia [20]. Nonetheless, both studies were influenced by confounding factors, as the concurrent use of fluid goal-directed therapy [19] or dexmedetomidine [20], which may have also impacted the outcomes. Conversely, studies by Aftab [23] and Honca [24] indicate no significant difference between TIVA and volatile anesthesia in the prevention of PONV. This absence of difference might be partly attributable to shorter durations of surgery compared to the aforementioned studies.

In a study by De Baerdemaeker et al., a comparison of the effect of volatile anesthetics in bariatric surgery, including sevoflurane and desflurane, revealed a higher incidence of PONV in the desflurane group at one time point, 120 min postoperatively [25]. The clinical significance of this single finding remains unclear and requires further studies.

Moreover, a single study assessing bispectral index (BIS)-guided monitoring of the depth of anesthesia using TIVA found no beneficial effects on PONV incidence [26], which contradicts studies in the general population [34].

#### 3.1.2. Neuromuscular Block

We identified three studies evaluating the impact of the depth of neuromuscular block and the reversal agent used on the incidence of PONV in bariatric patients. These two questions are connected, as the safe reversal of deep neuromuscular block implies a preference for sugammadex over neostigmine, which has a proven proemetic effect [35]. Such a relation is supported by a study by Castro et al., in which the sugammadex group had a lower incidence of PONV and an opioid-sparing effect compared to the neostigmine group [27].

With regard to deep neuromuscular blockade, consistent findings from two RCTs indicate that no reduction in PONV was observed [28,29]. Nonetheless, some advantageous effects on recovery, pain scores [28], or the restoration of intestinal function were noted [29]. In conclusion, the omission of neostigmine appears to be a crucial factor in mitigating the risk of PONV, and further research involving larger cohorts is necessary to evaluate the impact of neuromuscular blockade depth comprehensively.

#### 3.1.3. Perioperative Fluid Management

Proper hydration can reduce the incidence of PONV [36]. This can be accomplished by minimizing fasting duration and preoperative fluid restrictions, or by administering adequate fluid intra- and postoperatively, possibly using different methods to guide. In studies dedicated to bariatric surgery, an RCT by Suh et al. [30] demonstrated that providing patients scheduled for bariatric surgery with a carbohydrate drink up to three hours before surgery resulted in a reduced duration of PONV postoperatively, without adverse effects such as aspiration or disturbances in glycemic control.

Another approach to preventing both dehydration and excessive fluid administration involves the utilization of hemodynamic-guided goal-directed fluid therapy (GDFT), which has been evaluated in two studies [31,32]. Zheng et al., in their RCT, showed that GDFT based on stroke volume variation (SVV) was an effective method for maintaining the fluid balance, resulting in a lower incidence of PONV observed in the goal-directed group, 47.1% compared to 71.6% within the first 48 h [32]. Furthermore, the severity of symptoms and the requirement for rescue antiemetics were significantly reduced. Conversely, these promising findings are not supported by another study conducted by Cho et al. [31], which showed no significant benefit of SVV-guided fluid therapy concerning PONV in bariatric patients receiving crystalloids. Due to the limited research, more RCTs are warranted to assess the relationship and implications of various fluid administration strategies on the incidence of PONV.

#### 3.1.4. Nasogastric Tube Placement

The placement of a nasogastric tube is currently not recommended, not only in bariatric surgery [15], but also in other surgical procedures involving the upper gastrointestinal tract [37,38,39], as it fails to prevent complications, including PONV. Consistently, a single identified RCT dedicated to bariatric surgery demonstrated that nasogastric decompression with tube placement and maintenance in the postoperative period did not reduce the incidence of PONV or leakage in patients undergoing LSG [8]. To the contrary, nasogastric tube placement may hinder early oral intake, which is undesirable according to ERAS guidelines [15].

### 3.2. Opioid-Sparing Agents

The reduction in opioid dosage is a modifiable factor to decrease PONV incidence; therefore, opioid-sparing multimodal analgesia is a standard according to the ERAS guidelines [15]. However, the evidence supporting specific perioperative interventions varies, and in some instances, it may be limited. We identified 33 RCTs assessing the impact of opioid-sparing agents such as non-opioid analgesics, dexmedetomidine, lidocaine, N-methyl-D-aspartate (NMDA) receptor antagonists, ketamine and magnesium sulfate, gabapentinoids, and ultimately opioid-free anesthesia (OFA), which are detailed in Table 2. Dexamethasone is further included in pharmacological prophylaxis, although it also possesses co-analgesic properties.

**Table 2 jcm-14-06901-t002:** Opioid-sparing agents in bariatric surgery.

Author	Year	Number of Participants	Type of Surgery	Intervention Assessed	Dosing	Antiemetics Used	PONV Outcome Measure	Main Results Regarding PONV
Cooke [40]	2018	127	LSG	Paracetamol vs. placebo	Paracetamol 1000 mg i.v.	Aprepitant or droperidol, dexamethasone, ondansetron	11-point PONV scale until discharge	No difference between the groups
Amin [41]	2025	106	LSG, LYGB	Ketorolac vs. ibuprofen	Ketorolac 30 mg i.v. ibuprofen 800 mg i.v.	Dexamethasone, ondansetron	PONV incidence at an unspecified period	Ketorolac superior to ibuprofen
Govindarajan [42]	2005	50	LYGB	Ketorolac vs. remifentanil	Ketorolac infusion 6–9 mg/h i.v., postoperatively, infusion 4.5–9 mg/h i.v.	Ondansetron	PONV incidence at an unspecified period	PONV reduction 4% vs. 28% in the placebo group
Hamed [43]	2019	132	LSG	Dexmedetomidine vs. remifentanil	Dexmedetomidine 0.2–0.5 mcg/kg/h i.v. intraoperatively	Dexamethasone, droperidol	PONV incidence at an unspecified period	PONV reduction 3% vs. 10% in the placebo group
Ziemann-Gimmel [21]	2014	124	LSG, LYGB, LGB, revisional procedures	Propofol TIVA, OFA vs. volatile anesthesia with opioids	Dexmedetomidine loading dose 0.5 mcg/kg and infusion 0.1–0.3 mcg/kg/h i.v., ketamine 0.5 mg/kg i.v.	Dexamethasone, ondansetron, droperidol, promethazine	4-point Verbal Rating Scale in the morning postoperatively	PONV absolute risk reduction of 17.3% in the OFA group
Khail [44]	2023	90	LSG	Ketamine vs. Dexmedetomidine vs. placebo	Ketamine loading dose 0.3 mg/kg IBW, infusion 0.3/kg/h i.v., Dexmedetomidine loading dose 0.5 mcg/kg IBW, infusion 0.5 mcg/kg/h i.v.	Ondansetron	4-point PONV scale at 0, 6, 12, 24 h postoperatively	Less PONV in the dexmedetomidine group compared to ketamine and placebo
Abu-Halaweh [45]	2015	60	LBS	Dexmedetomidine vs. morphine	Dexmedetomidine 0.3 mcg/kg i.v.	Unspecified	PONV incidence within 24 h postoperatively	Less PONV in the dexmedetomidine group, 26.7% vs. 63.3%
Zeeni [46]	2019	60	LSG	Dexmedetomidine vs. morphine	Dexmedetomidine loading dose 1 mcg/kg, infusion 0.5 mcg/kg/h i.v.	Dexamethasone, ondansetron	NRS PONV score at 30 min intervals at PACU and 24 h postoperatively	No difference between the groups
Bakhamees [47]	2007	80	LYGB	Dexmedetomidine vs. placebo	Dexmedetomidine loading dose 0.8 mcg/kg, infusion 0.4 mcg/kg/h i.v.	Unspecified	PONV incidence within 24 h postoperatively	No difference between the groups
De Oliveira [48]	2014	50	LSG	Lidocaine vs. placebo	Lidocaine loading dose 1.5 mg/kg, infusion 2 mg/h i.v.	Ondansetron	PONV incidence at PACU	No difference between the groups
De Oliveira [49]	2020	60	LYGB	Lidocaine vs. placebo	Lidocaine loading dose 1.5 mg/kg, infusion 2 mg/h i.v.	Ondansetron	PONV incidence within 24 h postoperatively	Less PONV in the lidocaine group
Sheriff [50]	2017	150	LSG	Lidocaine vs. dexmedetomidine vs. placebo	Lidocaine loading dose 2 mg/kg, infusion 1.5 mg/kg/h i.v., dexmedetomidine loading dose 1 mcg/kg, infusion 0.4 mcg/kg/h i.v.	Unspecified	PONV incidence within 6 h postoperatively	No difference between the groups
Sun [51]	2022	99	LSG, LYGB	Lidocaine vs. TAP Block vs. placebo	Lidocaine loading dose 1.5 mg/kg, infusion 2 mg/kg/h i.v.	Dexamethasone	PONV incidence within 24 h postoperatively	No difference between lidocaine and TAP Block
Yurttas [52]	2023	137	LSG, LYGB	Lidocaine vs. placebo	Lidocaine loading dose 1.5 mg/kg, infusion 1.5 mg/kg/h i.v.	Unspecified	3-point PONV scale within 48 h postoperatively	No difference between the groups
Plass [53]	2020	178	LSG, LYGB	Lidocaine vs. placebo	Lidocaine loading dose 1.5 mg/kg, infusion 2 mg/kg/h i.v.	Dexamethasone, droperidol	PONV incidence until discharge	No difference between the groups
Sakata [54]	2020	58	LYGB	Lidocaine vs. placebo	Lidocaine loading dose 1.5 mg/kg, infusion 2 mg/kg/h i.v.	Ondansetron	PONV incidence until discharge from	No difference between the groups
Alimian [55]	2019	42	LYGB	Lidocaine 1 mg/kg/h vs. lidocaine 2 mg/kg/h	Lidocaine infusion 1 mg/kg/h vs. 2 mg/kg/h i.v.	Unspecified	PONV incidence at 0, 30 min, 1 h, 6 h, 12 h, 24 h postoperatively	No difference between the groups
Hasanein [56]	2011	60	LYGB	Ketamine and remifentanil vs. placebo	Ketamine 1 mcg/kg/min i.v.	Dexamethasone, metoclopramide	PONV incidence at an unspecified period	No difference between the groups
Yang [57]	2023	68	LSG	Esketamine	Esketamine loading dose 0.2 mg/kg, infusion 0.2 mg/kg/h i.v.	Dexamethasone	PONV score within 48 h postoperatively	No difference between the groups
Adhikary [58]	2021	126	LSG	Ketamine vs. Ketamine + Magnesium sulfate vs. placebo	Ketamine 0.5 mg/kg i.v., Magnesium sulfate 2 g i.v.	Dexamethasone, ondansetron	4-point PONV scale every 4 h until 24 h postoperatively	No difference between the groups
Mehta [59]	2020	54	LYGB	Ketamine vs. control	Ketamine loading dose 20 mg i.v., infusion 5 mcg/kg/min i.v.	Dexamethasone	Need for antiemetics within 24 h postoperatively	No difference between the groups
Zhang [60]	2023	74	LBS	Esketamine vs. placebo	Esketamine infusion 0.5 mcg/kg/h i.v.	Dexamethasone, Palonosetron	PONV incidence during the 1-day postoperatively	No difference between the groups
Schulmeyer [61]	2010	80	LSG	Pregabalin vs. placebo	Pregabalin 150 mg p.o.	Ondansetron	PONV incidence within 24 h postoperatively	Less PONV in the pregabalin group, 25.6% vs. 46.3% placebo
Alimian [62]	2012	60	LYGB	Pregabalin vs. placebo	Pregabalin 300 mg p.o.	Unspecified	PONV incidence within 24 h postoperatively	Absolute risk reductions of 36.7% for nausea and 30% for vomiting in the pregabalin group vs. placebo
Salama [63]	2016	60	LSG	Pregabalin and dexmedetomidine vs. placebo	Pregabalin 75 mg p.o., dexmedetomidine infusion 0.4 mcg/kg/h i.v.	Dexamethasone, ondansetron, metoclopramide	11-point Verbal Rating Scale every 30 min at PACU, every 2 h on the first post-surgery day	Significantly less PONV within the first 18 h postoperatively
Hassani [64]	2015	76	LYGB	Gabapentin vs. placebo	Gabapentin 100 mg p.o.	Unspecified	Incidence of PONV within 6 h postoperatively	Less PONV in the gabapentin group 10% vs. 33% placebo
Mulier [65]	2018	50	LSG, LYGB, revisional procedures	OFA vs. sufentanil	Dexmedetomidine loading dose 0.5 mcg/kg, infusion 0.25–1 mcg/kg/h i.v., ketamine loading dose 0.25 mg/kg i.v., lidocaine loading dose 1.5 mg/kg, infusion 1.5–3 mg/kg/h i.v.	Ondansetron	Incidence of PONV at PACU	PONV reduction 13% vs. 63% in the sufentanil group
Mieszczanski [66]	2023	59	LSG	OFA vs. remifentanil	Dexmedetomidine loading dose 1 mcg/kg, infusion max 1 mcg/kg/h i.v., ketamine loading dose 0.5 mg/kg i.v., lidocaine loading dose 1.5 mg/kg, infusion max 3 mg/kg/h i.v., magnesium sulfate 40–50 mg/kg i.v.	Dexamethasone, ondansetron	PONV Impact scale within 24 h postoperatively	Less PONV in the OFA group, limited to the first postoperative hour
Clanet [67]	2024	172	LYGB	OFA vs. remifentanil	Dexmedetomidine loading dose 0.5 mcg/kg, infusion 0.4–0.8 mcg/kg/h	Dexamethasone, ondansetron	Incidence of PONV at PACU and 4 h, 24 h postoperatively	Less PONV in the OFA group in the first 4 h after the surgery
Dagher [68]	2025	58	LSG, LYGB	OFA vs. opioid-based anesthesia	Dexmedetomidine infusion 0.2–0.5 mcg/kg/h i.v., Dexamethasone 8 mg i.v., Ketamine loading dose 0.2 mg/kg, infusion 0.15 mg/kg/h i.v., lidocaine loading dose 1.5 mg/kg, infusion 1.5 mg/kg/h i.v., magnesium sulfate loading dose 50 mg/kg, infusion 8 mg/kg/h i.v.	Ondansetron	Incidence of PONV 4 h, 24 h postoperatively	No difference between the groups
Menck [69]	2022	60	LYGB	OFA vs. opioid-based anesthesia	Dexmedetomidine loading dose 0.5 mcg/kg i.v., Ketamine 25 mg i.v., Lidocaine loading dose 1.5–2 mg/kg, infusion 1 mg/kg/h i.v., Magnesium sulfate loading dose 40 mg/kg i.v., infusion 5 mg/kg/h i.v., clonidine infusion 0.15 mcg/kg/h i.v.	Dexamethasone, droperidol, ondansetron	PONV incidence at PACU and on the first post-surgery day	No difference between the groups
Campos-Pérez [70]	2022	40	LYGB	OFA vs. opioid-based anesthesia	Dexmedetomidine loading dose 1–1.5 mcg/kg, infusion 0.3–0.7 mcg/kg/min i.v., ketamine loading dose 0.12 mg/kg, infusion 0.15 mcg/kg/min i.v., lidocaine 1 mg/kg, infusion 1 mg/kg i.v., magnesium sulfate loading dose 30–50 mg/kg, infusion 10 mg/kg/min i.v.	Unspecified	PONV incidence within 24 h postoperatively	No difference between the groups
Soudi [71]	2022	60	LBS	OFA vs. opioid-based anesthesia	Dexmedetomidine loading 1 mcg/kg, infusion 0.5 mcg/kg/h i.v., ketamine loading 0.5 mg/kg, infusion 0.25 mg/kg/h i.v.	Unspecified	Number of PONV episodes within 24 h postoperatively	No difference between the groups

#### 3.2.1. Non-Opioid Analgesics

The utilization of nonsteroidal anti-inflammatory drugs (NSAIDs) and paracetamol is advocated by both the ERAS [15] and Procedure Specific Postoperative Pain Management (PROSPECT) guidelines for LSG [72], based on data extrapolated from the broader obese population [73], as specific evidence regarding PONV incidence is scarce. Yet, 3 RCTs assessing various bariatric procedures consistently endorse the use of NSAIDs and paracetamol owing to their opioid-sparing effects, which are associated with a reduced incidence of PONV. In a study conducted by Cooke et al., patients undergoing LSG and receiving paracetamol experienced shorter hospital stays and a lower incidence of PONV; however, these differences did not achieve statistical significance, likely due to the limited sample size [40]. A recent study by Amin et al. demonstrated that ketorolac was highly effective, as its administration was correlated with a decrease in postoperative nausea from 87% to 42% and vomiting from 64% to 13%, compared to the ibuprofen group, alongside reductions in both intraoperative and postoperative opioid consumption [41]. These findings align with a study by Govindarajan et al., which demonstrated a reduction in PONV incidence from 28% to 4% in the ketorolac group compared to the placebo group [42], along with improved pain and recovery scores.

#### 3.2.2. Dexmedetomidine

Dexmedetomidine, an alpha-2 adrenergic agonist, has demonstrated a beneficial effect on the incidence of PONV in patients undergoing bariatric surgery [74] by decreasing the need for opioids and anesthetics, with most RCTs endorsing its protective role [21,43,44,45]. Hamed et al. compared dexmedetomidine infusion with remifentanil and reported a significantly lower incidence of PONV: 10% vs. 3% in the dexmedetomidine group, with better pain scores [43]. In this study, dexmedetomidine supplanted opioids intraoperatively without hemodynamic instability. In line with these results, Ziemann-Gimmel et al. demonstrated that fewer patients experienced PONV in the TIVA with propofol, dexmedetomidine, and ketamine group (20%) compared to 37.3% in the conventional, opioid-based anesthesia utilizing volatile anesthetics [21]. However, in their study, multiple interventions overlapped, making it difficult to estimate the isolated impact of dexmedetomidine. Furthermore, Khail et al. [44] in a three-arm study showed superiority in PONV reduction using dexmedetomidine over both the control group and the ketamine group and also improved recovery. Finally, Abu-Halaweh et al., in a single study comparing 24 h postoperative dexmedetomidine and morphine infusion, demonstrated a decreased incidence of nausea in the dexmedetomidine group (26.7% vs. 63.3%) [45]. Still, the morphine dosing regimen as a comparator was relatively high (3 mg/h) for LBS and more than standard in most of the centers.

In contrast to these RCTs, Zeeni et al. [46] found no difference between patients administered a loading dose (1 mcg/kg) of dexmedetomidine with a 30 min infusion and those receiving a placebo. Nevertheless, in this study, dexmedetomidine infusion was limited to a brief period during the surgical procedure, which may have attenuated its potential beneficial effects. Similarly, in a study conducted by Backhamees [47], no reduction in PONV was identified; however, their research demonstrated that dexmedetomidine enhanced pain scores, improved the recovery profile, and decreased opioid requirements relative to placebo.

#### 3.2.3. Lidocaine

The prevailing view recognizes intravenous administration of lidocaine, a sodium channel blocker, as a co-analgesic that improves recovery, decreases opioid consumption, and reduces the PONV incidence in patients undergoing LBS [75]. However, not only is the evidence not unequivocal and conflicting, but also concerns arise from the potential risk of bradycardia or hypotension and the risk of toxicity, especially if combined with different forms of regional analgesia.

In studies comparing lidocaine infusion with placebo, lidocaine superiority with regard to PONV reduction was shown in four studies [48,49,50,51]. In a study by Oliveira Jr., investigators observed improved recovery outcomes and a decreased incidence of PONV in Postoperative Care Unit (PACU) among patients who received a 1.5 mg/kg bolus and a 2 mg/kg/h infusion intraoperatively; however, due to a limited sample size of 25 patients per group, the difference did not reach statistical significance [48]. Notably, the same dose used in a study by Oliveira et al. was effective in preventing PONV for up to 24 h post-surgery [49]. In this latter study, the lidocaine group had both lower opioid consumption and lower pain scores during the immediate postoperative period. A study by Sherif et al., which examined various dosing regimens of lidocaine, including a bolus of 2 mg/kg and an infusion of 1.5 mg/kg/h, found that patients experienced less vomiting; however, the differences observed were not statistically significant [50]. Consistent with previous RCTs, Sun et al. demonstrated the beneficial effects of a lidocaine bolus of 1.5 mg IV combined with a 2 mg/kg/h infusion compared to placebo, by decreasing nausea from 63.63% in the control group to 42.42% in the lidocaine group, and reducing vomiting from 42.42% to 15.15%, respectively [51]. Interestingly, their study also indicated that lidocaine infusion was equally effective in reducing PONV and enhancing recovery as a transversus abdominis plane (TAP) block in obese patients undergoing LSG or LYGB.

Conversely, multiple investigations have cast doubts on the justification for administering lidocaine as an adjunct analgesic in bariatric surgery [52,53,54]. An RCT conducted by Yurttas et al., involving 137 patients, revealed no significant advantages of lidocaine administration concerning the incidence of PONV or opioid consumption within the first four hours postoperatively [52]. Unlike other studies, Yurttas et al. utilized lean body weight (LBW) for dosing adjustments, which may have resulted in a lower dosage compared to studies that employed adjusted body weight (ABW). Consistent with their findings, Plass et al. also failed to demonstrate the superiority of lidocaine infusion, finding no significant differences in PONV incidence, pain scores, or recovery metrics [53]. Furthermore, this study is among the few that examined potential hemodynamic instability in bariatric patients receiving perioperative lidocaine infusion; notably, refractory hypotension was observed in 6% of patients, whereas none were recorded in the control group. As a result, patients in the treatment group required increased vasopressor support, raising concerns regarding the safety profile of such a strategy. Lastly, Sakata et al. [54] reported no significant improvement in PONV incidence following lidocaine use; however, patients in the study group had significantly lower morphine consumption. It is essential to note that their study had a relatively small sample size (n = 58), which may have limited its statistical power to detect potential differences.

An important practical issue concerning the lidocaine dosing protocol in patients undergoing LBS is investigated by Alimian et al., who conducted a comparative study between a 1 mg/kg/h infusion and a 2 mg/kg/h infusion utilizing ideal body weight (IBW) [55]. The investigators found no significant differences in any of the assessed parameters, thereby concluding that lower dosages may be equally efficacious as higher dosages in managing postoperative pain and PONV within this patient population.

#### 3.2.4. NMDA Receptor Antagonists

##### Ketamine

Ketamine and its isomer S-ketamine are antagonists of NMDA receptors. Due to their analgesic and opioid-sparing properties, they are used as co-analgesics in bariatric surgery, exerting a beneficial influence on pain scores and opioid requirements [76]. Nevertheless, their impact on PONV prevalence remains controversial, as six identified RCTs have consistently shown no improvement in this aspect [44,56,57,58,59,60]. In a study conducted by Khail et al., patients receiving ketamine bolus and infusion demonstrated reductions in both pain scores and postoperative morphine consumption compared to placebo and dexmedetomidine groups [44]. Hasanein et al. [56] compared 60 patients assigned to either a remifentanil and ketamine infusion group or a placebo group. While this comparison revealed a substantial reduction in postoperative opioid consumption, which was potentially attributable, in part, to the attenuating effect of NMDA receptor antagonists on remifentanil-induced hyperalgesia [77], lower morphine consumption was not associated with a decreased incidence of PONV [56]. A similar relationship was observed in studies by Yang et al., Zhang et al., and Mehta et al. [57,59,60]. To some extent, this discrepancy may be explained by specific emetic properties of ketamine [78], which could balance the positive effects of opioid reduction.

Contrary to the aforementioned studies, Adhikary et al. [58], in their comparatively larger study involving 126 patients, did not demonstrate an opioid-sparing effect or enhanced pain scores, and consistently observed no difference in PONV. However, in contrast to other studies, patients received only a single bolus of ketamine rather than an infusion, which may have been a significant factor.

##### Magnesium Sulfate

Magnesium sulfate, also acting as an NMDA receptor antagonist, not only has opioid-sparing properties but also reduces PONV in the general population [79]. It is frequently incorporated as part of a multimodal drug regimen, including in bariatric surgery. Although the studies outlined in the OFA section utilized magnesium sulfate, only a single RCT evaluated its effect independently as a subgroup [58], and this study demonstrated no significant beneficial outcomes. Nevertheless, in that study, the dosage of magnesium sulfate was relatively low (2 g) and administered regardless of patient weight, potentially leading to underdosing in obese patients and, consequently, reduced efficacy. Given the limited data available, further RCTs are required to comprehensively evaluate the impact of magnesium sulfate on PONV in bariatric surgical procedures.

#### 3.2.5. Gabapentinoids: Gabapentin and Pregabalin

Gabapentin and pregabalin are derivatives of the neurotransmitter gamma-aminobutyric acid (GABA), which bind to the α2δ subunit of the voltage-gated calcium channel, thereby inhibiting excitatory synaptic transmission. In addition to their use as anticonvulsants or for the treatment of neuropathic pain, these medications may be administered as co-analgesics during various surgical procedures, including bariatric surgery, to enhance pain control and reduce PONV [80]. Nevertheless, gabapentinoids represent the least studied co-analgesics used in LBS, with only four studies evaluating their impact on PONV and one study in progress [61,62,63,64,81].

Schulmeyer et al. demonstrated a beneficial impact of a pregabalin dose of 150 mg compared to placebo regarding PONV incidence and postoperative pain [61] in their study of 80 enrolled patients. The pregabalin group had a lower incidence of PONV at 25.6% (10/39) compared to 46.3% in the placebo group (19/41), respectively, resulting in an absolute risk reduction of approximately 20%, which is clinically significant. Additionally, morphine use and pain scores were superior to those of the placebo. Moreover, the pregabalin group experienced no apparent side effects, such as excessive sedation, which would impair recovery and diminish the mentioned benefits [61]. However, a critical limitation of this study is the assumption that multimodal analgesia or simple analgesics are not used, restricting pregabalin’s administration to cases where these alternatives are not feasible, which is mainly hypothetical [72]. Moreover, Alimian et al. found that 300 mg of pregabalin reduced the risk of PONV compared to placebo [62], achieving absolute risk reductions of 36.7% for nausea and 30% for vomiting. The study’s shortcoming is the lack of reporting on the incidence of oversedation and delayed recovery, as pregabalin, especially at higher doses, can have a sedative effect that should be avoided in the ERAS protocol [15]. In line with these results, Salama et al. reported a significant reduction in PONV in patients receiving 75 mg of pregabalin with dexmedetomidine infusion [63] up to 18 h after the LSG. However, in their study, the potentially beneficial impact of pregabalin overlaps with that of dexmedetomidine, therefore making it difficult to determine the effect of pregabalin as a sole co-analgesic. Finally, PONV risk reduction is also supported by Hassani et al. [64], who reported a 10% vs. 33.3% PONV incidence in a group that received gabapentin 100 mg. The demonstrated efficiency of even such a small dose of gabapentinoids in PONV prophylaxis is promising, and it should entail larger, multi-center dose-finding studies to fully assess their potential in improving post-operative outcomes in LBS.

#### 3.2.6. Opioid-Free Anesthesia (OFA)

OFA is characterized as a heterogeneous group of techniques wherein opioids are not administered during general anesthesia [82]. OFA encompasses comprehensive multimodal analgesia, advancing from opioid-sparing strategies to the complete elimination of intraoperative opioids. Due to variations in dosing regimens and the potential application of regional anesthesia techniques across studies, RCTs evaluating this method are heterogeneous, and comparing them yields conflicting results. Yet, concerning PONV prophylaxis, the recent network meta-analyses demonstrate OFA superiority compared to opioid-based and opioid-sparing anesthesia, with an overall relative risk reduction of 40% [83]. Conversely, although the prevailing consensus regards OFA as a protective measure against PONV, the number of RCTs is limited. There are significant discrepancies among these studies, and debates persist concerning the specific time points at which the incidence and severity of PONV are evaluated. Furthermore, OFA raises concerns regarding potential hemodynamic instability and patient safety, which warrant careful consideration [84].

In their pivotal study, Mulier et al. demonstrated a statistically significant PONV reduction from 63% in the opioid-based group using sufentanil to 13% in the OFA group, measured during the immediate postoperative period in PACU [65]. The benefits of OFA also encompassed reduced morphine consumption and improved recovery outcomes. However, the authors did not assess PONV in the following hours, and it could not be concluded whether the antiemetic protective effect was prolonged. Aligning with these findings, a study by Mieszczanski et al. reported that the superiority of OFA concerning PONV incidence was observed only during the first hour post-surgery, with no significant difference from the sixth hour onwards. The authors attributed this pattern to the short half-lives of the co-analgesics administered exclusively during surgery [66]. Supporting these observations, Clanet et al. noted a reduction in PONV within the first four hours postoperatively in the OFA group [67]. Nevertheless, in a previous study by Ziemann-Gimmel et al. [21], this beneficial effect persisted up to 24 h in patients receiving OFA with propofol TIVA, rendering it more clinically significant. Conversely, several studies have failed to demonstrate any improvement in PONV associated with the use of OFA [68,69,70,71]. These discrepancies, together with the absence of definitive evidence, contribute to the ongoing debate regarding the role of OFA in bariatric surgical procedures.

### 3.3. Pharmacological Prophylaxis

Given that PONV prophylaxis using a single drug in bariatric surgery may prove insufficient, current ERAS guidelines advocate for a multimodal approach [15]. This strategy involves the administration of one antiemetic agent from at least three of the following six classes of drugs: long-acting corticosteroids, such as dexamethasone; 5-hydroxytryptamine receptor antagonists (5-HT); butyrophenones; neurokinin-1 (NK-1) receptor antagonists; antihistamines; and anticholinergics [15]. We identified 16 RCTs evaluating such antiemetics listed in Table 3.

**Table 3 jcm-14-06901-t003:** Pharmacological PONV prophylaxis.

Author	Year	Number of Participants	Type of Surgery	Intervention Assessed	Dosing Regimen	Antiemetics Used	PONV Outcome Measure	Main Results Regarding PONV
Didehvar [85]	2013	76	LBS	Dexamethasone vs. placebo	Dexamethasone 8 mg i.v.	Palonosetron	4-point PONV scale at 2, 6, 12, 24, 72 h postoperatively	No difference between the groups
Benevides [86]	2013	90	LSG	Ondansetron vs. dexamethasone + ondansetron + haloperidol	Dexamethasone 8 mg i.v., Ondansetron 8 mg i.v., Haloperidol 2 mg i.v.	Unspecified	11-point Verbal Rating Scale at 0–2, 2–12, 12–24, 24–36 h postoperatively	Less nausea in the dexamethasone, haloperidol, and dexamethasone group compared to the ondansetron-only group
Spaniolas [22]	2020	83	LSG	Propofol TIVA, aprepitant, and transdermal scopolamine vs. volatile anesthesia	Aprepitant 80 mg p.o., Dexamethasone 8 mg i.v., ondansetron 4 mg iv., scopolamine transdermal patch	Metoclopramide, compazine	10-point Verbal Rating Scale at 1, 4, 12, 24 h postoperatively	PONV scores lower at all the time points in the study group
Bataille [4]	2016	117	LSG	Dexamethasone and ondansetron vs. placebo	Dexamethasone 4 mg i.v., ondansetron 4 mg i.v.	TIVA	PONV incidence within 24 h postoperatively and 11-point Verbal Rating Scale	Less PONV in the study group 45% vs. 54% in the control group
Burcu [87]	2024	100	LSG	Ondansetron vs. Palonosetron	Ondansetron 8 mg i.v., Palonosetron 75 mcg i.v.	Unspecified	PONV incidence during hospitalization	Less PONV in the palonosetron group
Kaloria [88]	2017	22	LSG	Ondansetron vs. palonosetron	Ondansetron max 8 mg i.v., palonosetron max 75 mcg	Dexamethasone	PONV incidence at 0, 0–6, 6–12, 12–24, 24–48, 48–72 h postoperatively	No difference between the groups
Ortiz [89]	2024	400	LSG	Aprepitant vs. placebo	Aprepitant 80 mg p.o.	Dexamethasone, ondansetron, metoclopramide	Rhodes index at 0, 6, 12, 24 h postoperatively	Less PONV in the aprepitant group in the first 24 h postoperatively
Sinha [90]	2014	125	LYGB, LGB	Aprepitant vs. placebo	Aprepitant 80 mg p.o.	Ondansetron	PONV incidence at 30 min, 1, 2, 6, 24, 48, 72 h postoperatively	Less vomiting in the aprepitant group, 3% vs. 15% in the placebo group
Ashoor [91]	2022	90	LSG	Aprepitant vs. mirtazapine vs. placebo	Aprepitant 80 mg p.o., mirtazapine 30 mg p.o.	Dexamethasone	4-point Verbal Descriptive Scale every 4 h within 24 h postoperatively	Less PONV in the aprepitant group 34.5% vs. mirtazapine, 35.7% and placebo, 93.1%
Shahinpour [92]	2024	83	LSG, LYGB	Haloperidol vs. promethazine	Haloperidol 2 mg i.m., promethazine 25 mg i.m.	Dexamethasone, ondansetron	Numeric Verbal Rating Scale at 6 and 24 h postoperatively	Less PONV in the haloperidol group, 20% vs. promethazine 40%
Talebpour [93]	2017	80	Laparoscopic gastric plication	Metoclopramide vs. promethazine	Metoclopramide 10 mg i.v., promethazine 50 mg i.m.	Dexamethasone, ondansetron	4-point PONV scale within 48 h postoperatively	Less PONV in the promethazine group 41% vs. 97.5% in the metoclopramide group
Moussa [94]	2007	120	LSG, LYGB, LGB	Granisetron vs. granisetron + droperidol vs. granisetron + dexamethasone vs. placebo	Dexamethasone 8 mg i.v., droperidol 1.25 mg i.v., granisetron 1 mg i.v.	Unspecified	PONV incidence within 24 h postoperatively	PONV incidence 30% in granisetron, 30% granisetron + droperidol, 20% granisetron + dexamethasone vs. 67% in the placebo group
Ebrahimian [95]	2023	130	LSG	Ondansetron vs. metoclopramide vs. granisetron vs. metoclopramide + ondansetron	Ondansetron max 8 mg i.v., metoclopramide max 10 mg i.v., granisetron 2 mg i.v.	Dexamethasone	PONV impact scale within 48 h postoperatively	No difference between the groups
Pourfakhr [96]	2021	82	LSG	Diphenhydramine vs. control	Diphenhydramine 0.4 mg/kg i.v.	Ondansetron	PONV incidence within 24 h postoperatively	Less PONV in the diphendydramine group 30% vs. 56% in the control group
Ahmed [97]	2025	210	LSG	Cyclizine vs. metoclopramide vs. ondansetron	Cyclizine 50 mg i.v., metoclopramide 10 mg i.v., ondansetron 8 mg i.v.	Dexamethasone	11-point Likert scale within 24 h postoperatively	No difference between the groups
Atif [98]	2022	100	LSG	Scopolamine vs. control	Scopolamine 10 mg i.v.	Metoclopramide	PONV intensity scale at 2 and 24 h postoperatively	No difference between the groups

#### 3.3.1. Dexamethasone

Dexamethasone is an established antiemetic agent used to decrease PONV incidence [36] and is endorsed for routine application in bariatric surgery by the American Society for Metabolic and Bariatric Surgery (ASMBS) [99]. It is regarded as particularly advantageous in laparoscopic surgeries, as it may enhance pain scores and decrease opioid consumption [15]. Consequently, it is incorporated into most multi-drug prophylactic regimens against PONV in laparoscopic bariatric surgery (LBS), given that its single-dose benefits outweigh potential risks, such as transient hyperglycemia [100]. Nonetheless, the number of RCTs dedicated explicitly to assessing its isolated efficacy in preventing PONV within this context remains limited. Didehvar et al. reported in their study that the addition of dexamethasone to the 5-HT3 antagonist palonosetron was ineffective in reducing PONV risk among high-risk bariatric surgery patients [85]. Consistent with these findings, Benevides et al. [86] observed no statistically significant difference in PONV incidence between the groups receiving ondansetron alone, ondansetron combined with dexamethasone, and ondansetron with haloperidol, respectively. However, their study noted that groups administered dexamethasone experienced lower pain scores and reduced postoperative opioid consumption.

Dexamethasone is more frequently evaluated as part of a multimodal therapeutic approach. In such a study, Spaniolas et al. demonstrated that dexamethasone, combined with aprepitant, transdermal scopolamine, and ondansetron, significantly improved verbal scores related to nausea and vomiting at all measured time points compared to the control group [22]. Of clinical importance is the elimination of discharge delays associated with PONV, contrasting with the 9.5% incidence observed in the control group. In contrast to these results, and highlighting the difficulty of preventing PONV among high-risk bariatric patients, an RCT conducted by Bataille et al. revealed that the combination of dexamethasone and ondansetron did not effectively prevent PONV in patients undergoing laparoscopic sleeve gastrectomy (LSG) [4]. It is noteworthy that the dose of dexamethasone used in this study was relatively low (4 mg), which may have contributed to the lack of efficacy of this prophylactic strategy.

#### 3.3.2. 5-HT3 Antagonists

5-HT3 antagonists, such as ondansetron, palonosetron, or granisetron, are recommended for both prophylaxis and rescue treatment of PONV and are regarded as the gold standard for these applications [36,101]. Consequently, they are frequently incorporated into combination therapy with antiemetics of different classes. Although they are considered first-line treatments, there exists limited evidence supporting the superiority of one particular agent over others. Burcu et al. demonstrated the superiority of palonosetron, administered at a dose of 1 mcg/kg of total body weight, in comparison to ondansetron, given at a fixed standard dose of 8 mg intravenously [87]. In their RCT, palonosetron was associated with a lower incidence of PONV during the first 24 h. Furthermore, only 2% of patients in the study group required rescue antiemetics, compared to 68% of patients who received ondansetron, marking a significant difference. The authors attribute this discrepancy to the longer duration of action of palonosetron, its weight-adjusted dosing regimen, and enhanced efficacy in preventing PONV [87]. Conversely, a study conducted by Kaloria et al. [88] found that the difference in PONV incidence between the 8 mg ondansetron group and the 75 mcg palonosetron group, both with dexamethasone, was too minor to attain statistical significance. However, the study’s limited sample size, as it was designed as a pilot study, indicates the need for larger RCTs.

#### 3.3.3. NK-1 Receptor Antagonists

RCTs conducted in patients undergoing bariatric surgery have demonstrated that aprepitant, an NK-1 receptor antagonist, is a promising and potent agent for preventing PONV [102], particularly since it has no sedative effects. In a recent meta-analysis, patients receiving aprepitant had a lower incidence of PONV, at 17.7% compared to 30.7% in the control group [102]. Furthermore, the protective effect was sustained for up to 12 h postoperatively following a single dose of aprepitant. An extensive study involving 400 participants, conducted by Ortiz et al., revealed that administering 80 mg of aprepitant one hour preoperatively, in addition to a multimodal prophylaxis regimen comprising dexamethasone, ondansetron, and metoclopramide, significantly reduced nausea, vomiting, and the need for rescue medication up to 24 h after the LSG [89]. Similarly, Sinha et al. reported a reduction in the incidence of postoperative vomiting, along with a delay in the onset of the first vomiting episode [90], following the administration of 80 mg of aprepitant one hour before the surgery. However, their study found no statistically significant differences in the mean nausea verbal score. Aligning with the aforementioned RCTs, Ashoor et al. demonstrated a superior antiemetic effect of 80 mg of aprepitant combined with 8 mg of dexamethasone compared to dexamethasone alone or mirtazapine [91], revealing a PONV incidence of 34.5% in the aprepitant group versus 93.1% in the dexamethasone-alone group. Additionally, the authors reported that patients in the aprepitant group achieved the highest satisfaction scores, which constitutes a clinically significant finding.

Considering the increasing popularity of aprepitant, a significant knowledge gap remains: the optimal dosage has yet to be identified. This matter is of particular importance, as the standard dose recommended by the European Society of Anesthesiology and Intensive Care (ESAIC) guidelines is 40 mg [36]; 80 mg is the most extensively studied in the LBS, and 125 mg is also evaluated in some studies [103]. Moreover, to our knowledge, no RCTs dedicated to bariatric surgery have evaluated other NK-1 antagonists such as fosaprepitant and their efficacy in preventing PONV, which would be of considerable interest given their proven effectiveness in gastrointestinal surgery [104].

#### 3.3.4. Dopamine Receptor Antagonists

This drug class encompasses dopamine receptor antagonists, including antipsychotics such as butyrophenones, derivatives of phenothiazine, and the prokinetic agent metoclopramide. Butyrophenones, exemplified by droperidol and haloperidol, possess established antiemetic properties and have been used as first-line PONV prophylaxis. Yet, their usage has diminished due to concerns regarding their safety profile and sedative effects, which are considered undesirable within the ERAS protocol [15]. These disadvantages must be carefully considered, rendering this class of drugs more suitable as a rescue therapy rather than a first-line treatment [36].

In a study conducted by Shahinpour et al., a multi-drug prophylactic regimen comprising dexamethasone, ondansetron, and haloperidol demonstrated greater efficacy in preventing PONV compared to promethazine [92]. The study reported a twofold reduction in PONV incidence within the haloperidol group, with 20% of patients experiencing PONV versus 40% in the control group during recovery among those undergoing bariatric surgery. Furthermore, both the incidence and severity of PONV were diminished following haloperidol administration. Nevertheless, the authors did not evaluate the sedation levels or the time to ambulation in the patients, which raises concerns regarding whether the benefits of the observed effect surpass potential risks [92].

Promethazine, alternatively, possesses both antipsychotic and antihistaminic properties, with a dose-dependent sedative effect [36,105]. Talebpour et al. observed that promethazine was more effective than metoclopramide in preventing PONV, reducing the incidence of PONV by 41% in comparison to 95.5% [93]. Nonetheless, oversedation occurred more frequently in the promethazine group, with a statistically significant difference observed in the mean duration of ambulation during the post-operative period [93], which constitutes a profound adverse effect given that early ambulation is a priority for this patient population.

On the other hand, Moussa et al. [94] demonstrated that the combination of haloperidol with granisetron did not produce a statistically or clinically significant effect on the incidence of PONV among patients undergoing LBS, thereby questioning its role within a multi-drug antiemetic regimen.

Although metoclopramide seems to have low antiemetic efficacy with a number needed to treat (NNT) of 8–10 [106], an RCT by Ebrahimian et al. demonstrated a synergistic effect of the combination of ondansetron and metoclopramide, exceeding the efficacy of monotherapy with metoclopramide, ondansetron, or granisetron [95]. However, the sample size was relatively small, requiring larger RCTs to confirm this effect.

#### 3.3.5. Histamine Receptor Antagonists

Antihistamines such as dimenhydrinate have demonstrated efficacy in preventing PONV [107]. However, there is a limited number of studies on the bariatric population, and concerns remain regarding their sedative effects, which restrict their use [36]. Addressing these issues, Pourfakhr et al. reported that patients undergoing LSG who received diphenhydramine, in comparison to those receiving a placebo, experienced a significantly lower incidence of PONV (30% versus 56% in PACU, and 40% versus 66% within 24 h), as well as improved pain scores [96]. While increased sedation scores indicated a sedative effect of diphenhydramine, this was not associated with prolonged PACU stays or delayed recovery, which holds clinical significance [96].

Conversely, Ahmed et al. observed no significant difference in the incidence of PONV between the groups administered dexamethasone and those receiving histamine receptor antagonists such as cyclizine, metoclopramide, or ondansetron [97], considering these interventions to be equally efficacious in preventing PONV without any adverse effects attributable to the medications.

#### 3.3.6. Muscarinic Receptor Antagonists

Scopolamine is an anticholinergic medication that, when used as a long-acting transdermal patch, possesses the potential to reduce the incidence of PONV during the first 24 h following surgery and potentially beyond [108]. The primary adverse effects associated with this strategy include xerostomia, blurred vision, and sedative effects. In one RCT dedicated to bariatric surgery, Atif et al. observed no advantage of intravenous scopolamine administration before stapling the gastric sleeve regarding PONV incidence [98] in patients undergoing LSG. Similarly, in one study involving major laparoscopic surgeries, including bariatric procedures, limited benefits from the transdermal application of scopolamine were demonstrated [109]. On the other hand, Spaniolas et al. [22] successfully integrated the transdermal scopolamine patch into multimodal prophylaxis for PONV; however, due to the study design, the efficacy of this technique cannot be evaluated independently.

In conclusion, the role of scopolamine in preventing PONV in bariatric surgery remains unclear; however, the extended duration of action associated with the transdermal patch may be advantageous and warrants further investigation.

### 3.4. Regional Anesthesia

Current evidence consistently recommends utilizing regional anesthesia techniques in obese patients undergoing bariatric surgery, as they reduce opioid use and, therefore, decrease the risk of PONV [15,110]. Such methods include local wound infiltration, intraperitoneal administration of local anesthetics, TAP Block, Erectus Spinae Plane Block (ESPB), Quadratus Lumborum Block (QLB), and External Oblique Intercostal Fascial Plane Block (EOIB). We identified 19 such studies, which are listed in Table 4.

**Table 4 jcm-14-06901-t004:** Regional Anesthesia in Bariatric Surgery.

Author	Year	Number of Participants	Type of Surgery	Intervention Assessed	Control group	Antiemetics Used	PONV Outcome Measure	Main Results Regarding PONV
Shariat [111]	2025	40	LSG	QLB	Wound infiltration with local anesthetics	Ondansetron	PONV incidence within 24 h postoperatively	No difference between the groups
Ma [112]	2019	179	LYGB	Wound infiltration with liposomal bupivacaine	Wound infiltration with bupivacaine alone	Dexamethasone, transdermal scopolamine	PONV score every 4 h until discharge	No difference between the groups
Neishaboury [113]	2025	72	LSG	Intraperitoneal ropivacaine alone vs. intraperitoneal ropivacaine with dexmedetomidine	Intraperitoneal placebo	Ondansetron	PONV incidence within 24 h postoperatively	Less PONV in the intraperitoneal ropivacaine with dexmedetomidine 8.3% vs. ropivacaine alone 29.2% vs. 50% placebo
Alamdari [114]	2018	120	LSG	Intraperitoneal bupivacaine	Control group	Unspecified	PONV incidence at unspecified time	Less PONV in the bupivacaine group 11.7% vs. 41.7% control
Omar [115]	2019	100	LBS	Intraperitoneal bupivacaine	Intraperitoneal placebo	Dexamethasone, ondansetron	PONV incidence within 24 h postoperatively	No difference between the groups
Kaur [116]	2022	104	LBS	Intraperitoneal ropivacaine	Intraperitoneal placebo	Dexamethasone, droperidol, ondansetron	Antiemetic use at 1, 2, 4, 6, 24, 48 h postoperatively	No difference between the groups
Sherwinter [117]	2008	30	Laparoscopic adjustable banding	Intraperitoneal bupivacaine	Intraperitoneal placebo	Dexamethasone, metoclopramide	PONV incidence at 30 min, 6, 12, 24, 48 h postoperatively	No difference between the groups
Mittal [118]	2018	60	LSG	TAP Block	No intervention	Unspecified	PONV incidence at 30 min, 3, 6, 12, 24, 48 h postoperatively	Less PONV in the TAP Block group
Emile [119]	2019	92	LBS	TAP Block	No intervention	Ondansetron	4-point PONV scale within 24 h postoperatively	Less PONV in the TAP Block group
Ibrahim [120]	2014	63	LSG	Subcostal TAP Block	Wound infiltration with local anesthetics	Dexamethasone, ondansetron	PONV incidence at PACU and at 24 h postoperatively	No difference between the groups
Zhou [121]	2024	71	LSG	TAP Block	No intervention	Dexamethasone, ondansetron, droperidol	PONV score at 2, 6, 12, 24 h postoperatively	Less PONV in the TAP Block group
Albrecht [122]	2013	70	LYGB	Subcostal TAP Block + wound infiltration with local anesthetic	Wound infiltration with local anesthetic	Dexamethasone, ondansetron, granisetron	PONV incidence during phase I recovery, 0–24, 24–48 h postoperatively	No difference between the groups
Saber [123]	2019	90	LSG	TAP Block with bupivacaine, TAP Block with bupivacaine + adrenaline	No intervention	Dexamethasone, metoclopramide	5-point PONV score every 6 h for 48 h	No difference between the groups
Karaveli [124]	2025	40	LSG	ESP Block	No intervention	Ondansetron	4-point Likert scale within 24 h postoperatively	No difference between the groups
Mostafa [125]	2021	60	LBS	ESP Block	Sham block	Dexamethasone	PONV incidence within 24 h postoperatively	No difference between the groups
Omran [126]	2021	30	LSG, LYGB	QLB	Sham block	Unspecified	PONV incidence within 24 h postoperatively	Less PONV in the QLB group 13.3% vs. 46.7%
Xue [127]	2022	225	LSG	TAP Block, QLB	No intervention	Granisetron	PONV incidence	No difference between the groups
Ozel [128]	2025	60	LSG	EOIB	No intervention	Dexamethasone, ondansetron	5-point Verbal Descriptive Scale within 24 h postoperatively	Fewer patients requiring antiemetics in the EOIB group, 16.7% vs. 40%
Turunc [129]	2024	58	LSG	EOIB	M-TAPA Block	Ondansetron	5-point Verbal Descriptive Scale at 0, 3, 6, 12, 18, 24 h postoperatively	No difference between the groups

#### 3.4.1. Local Wound Infiltration

Local wound and trocar insertion site infiltration is regarded as a practical alternative to regional blocks owing to its simplicity [110]. However, the research concerning the incidence is minimal. In an RCT by Shariat et al., the authors found no difference regarding PONV incidence comparing local anesthetic wound infiltration to posterior QLB in patients undergoing LSG [111], thereby indicating the efficacy of the former method. Additionally, the type of local anesthetic employed does not appear to have a definitive impact [112].

#### 3.4.2. Intraperitoneal Local Anesthetic Installation

Although the intraperitoneal administration of local anesthetics is currently not recommended in bariatric surgery due to ongoing controversies [72], emerging evidence suggests potential benefits, including an improvement in the incidence. Neishaboury et al. observed that intraperitoneal administration of 0.5% ropivacaine with 1 mcg/kg dexmedetomidine resulted in a significantly reduced incidence of PONV within 24 h, compared to both placebo (8.3% versus 50%) and ropivacaine alone (29.2%) [113]. The precise mechanism, whether attributable to local or systemic effects of dexmedetomidine, remains a topic of debate. Consistent with these findings, Alamdari et al. reported a lower incidence of PONV in patients receiving intraperitoneal bupivacaine (11.7% versus 41.7%) [114]. On the other hand, other studies have not demonstrated any improvement in PONV incidence [115,116,117], thereby casting doubt on the overall efficacy of this approach. Although further large-scale RCTs are necessary to comprehensively evaluate this technique, alternative methods supported by more unequivocal evidence may be preferred.

#### 3.4.3. TAP Block

The TAP block is a regional anesthetic technique that provides analgesia to the anterior and lateral abdominal wall [130]. Two RCTs by Emile and Mittal demonstrated the superiority of the TAP block in comparison to control groups [118,119]. In these studies, it was consistently observed that not only was the incidence and severity decreased in the TAP group, but also the time to ambulation, pain scores, and overall patient satisfaction [118,119]. Similarly, a study evaluating the subcostal TAP block compared to local anesthetic infiltration by Ibrahim et al. noted a marginal reduction in nausea and vomiting among patients receiving TAP blocks [120]. Zhou reported a decrease in PONV in patients who received TAP blocks with dexmedetomidine and esketamine, compared to those under opioid-based anesthesia. However, the benefit of regional anesthesia in this study is difficult to ascertain due to numerous confounding variables between groups [121]. In contrast, two randomized trials conducted by Albrecht and Saber found no significant differences in the rates of nausea or vomiting between the experimental and control groups [122,123]. Considering that only some of the RCTs evaluating TAP block in bariatric surgery assess PONV, and given their small sample sizes, the overall impact of TAP block, specifically on PONV incidence, is subject to debate and necessitates further randomized controlled trials, particularly comparisons with other regional anesthesia techniques. Nonetheless, the use of the TAP block as part of multimodal analgesia presents undeniable benefits, including a significant reduction in opioid consumption [131].

#### 3.4.4. ESPB

ESPB is a fascial plane block wherein local anesthetic is administered between the erector spinae muscle and the thoracic transverse process [132]. The main objective is to inhibit the dorsal and ventral branches of the thoracic and abdominal nerves. In an RCT by Karaveli et al., it was demonstrated that bilateral ESPB significantly diminishes analgesic consumption; however, the sample size was too small to determine its effect on PONV [124]. Similar findings and limitations were reported in a study by Mostafa et al. [125]. In contrast, a randomized trial by Jinaworn et al. revealed no significant difference in perioperative opioid consumption in patients undergoing laparoscopic bariatric surgery (LBS). The authors suggested that this may have been attributable to routine infiltration of local anesthetic by surgeons in both the control and experimental groups, as well as the routine administration of other analgesics [133]. This study, however, did not report on PONV incidence. In conclusion, ESPB provides superior pain control compared to placebo in patients undergoing metabolic surgery and appears to decrease the risk of PONV, likely due to reduced intraoperative and postoperative opioid consumption [134].

#### 3.4.5. QLB

One of the other regional anesthesia techniques is the QLB, wherein the needle tip is positioned between the quadratus lumborum muscle and the psoas major muscle for the administration of local anesthetic. Omran et al. demonstrated that bilateral posterior QLB significantly reduced the incidence from 46.7% in the control group to 13.3% in the QLB group, with improved pain scores and decreased opioid consumption [126]. On the other hand, Xue et al. demonstrated that despite providing better analgesia in TAPB and QLB groups, no corresponding improvement in PONV was achieved [127].

#### 3.4.6. EOIB

EOIB is a novel technique that may reduce pain by blocking both the lateral and anterior cutaneous branches of the intercostal nerves. It may offer benefits, as it is easier to perform due to the superficial location of the fascial plane. We identified only 2 RCTs evaluating this technique in bariatric surgery. Ozel et al. demonstrated that it provides adequate analgesia in patients undergoing LSG and reduces PONV scores compared to the control group, with fewer patients requiring antiemetics 16.7% vs. 40% [128]. Another option is performing a Modified Thoracoabdominal Nerve Block Through Perichondrial Approach (M-TAPA), which has effects similar to EOIB, as demonstrated by Turunc et al. [129], and shows no significant differences in PONV incidence or pain scores. However, the number of studies investigating these novel blocks remains limited.

## 4. Discussion

This scoping review provides a synthesis of current evidence based on RCTs concerning perioperative methods aimed at reducing the risk and incidence of PONV in patients undergoing bariatric surgery. Although the included studies demonstrate a broad spectrum of potential preventative strategies, PONV prophylaxis remains a significant challenge and a prevalent issue among a considerable proportion of patients [1]. A key conclusion derived from this review is that PONV prophylaxis, particularly in this vulnerable group, must be multimodal, integrating diverse mechanisms of action, as no single intervention is sufficient to ensure patient comfort in this regard.

The factors influencing PONV discussed in this review include anesthesia techniques, opioid-sparing strategies such as the use of co-analgesics, pharmacological prophylaxis, and regional anesthesia.

In patients undergoing bariatric surgery, despite the practical utility and potential benefits of volatile anesthetics, propofol TIVA is deemed superior for the prevention of PONV [19,20,22,34,35]. Regarding muscle relaxation, although the effectiveness of deep neuromuscular blockade in preventing PONV remains uncertain, sugammadex should be preferred over neostigmine to mitigate the associated risks [27]. Furthermore, avoiding dehydration through the oral administration of carbohydrate fluids and ensuring adequate intravenous fluid therapy is recommended [30,31]. However, the application of hemodynamic-guided fluid therapy produces conflicting outcomes and necessitates further research [31,32], as it is not yet regarded as a standard treatment.

Given that opioid utilization is among the primary modifiable factors contributing to PONV, the implementation of opioid-sparing strategies within multimodal analgesia is unequivocally endorsed [15]. Besides the consensus that NSAIDs and paracetamol possess opioid-sparing properties, thereby reducing the likelihood of PONV [135], the selection of co-analgesics remains less definitive. Although accumulating evidence supports the administration of dexmedetomidine [21,43,44,45,74] to decrease the incidence of PONV, enhance pain management and recovery outcomes, some studies question its efficacy in preventing PONV and highlight potential adverse effects [46,47,84]. Similarly, various studies have demonstrated that lidocaine, as a co-analgesic, can decrease PONV [48,49,50,51]; however, controversies persist concerning optimal dosing regimens, safety profiles, and the combination with regional anesthesia modalities, each involving local anesthetics and presenting a potential risk of toxicity. Furthermore, conflicting results have emerged regarding lidocaine’s effectiveness and the duration of its PONV-reducing effects [52,53,54]. On the other hand, ketamine, known for enhancing pain scores and reducing opioid requirements [76], does not appear to significantly lower PONV risk [44,47,56,57,58,59], possibly due to its emetogenic properties [78]. Magnesium sulfate, although promising, currently lacks sufficient evidence to support its use for PONV mitigation [58]. Lastly, gabapentinoids are considered potentially advantageous in reducing PONV risk [61,62,63,64]; however, the optimal dosing regimen remains unresolved.

The elimination of opioids through the utilization of OFA may represent a significant, evidence-based strategy to mitigate the risk of PONV, which is supported by the increasing amount of literature [21,65,66,83]. However, as OFA is a highly heterogeneous group of techniques, differing in dosing regimens and the selection of co-analgesics, no specific model can be recommended. Moreover, as there are concerns about its safety, progressing from multimodal opioid-sparing anesthesia to OFA is not currently regarded as standard practice.

Pharmacological multi-drug prophylaxis for PONV remains a standard approach [15,36]. Given its concomitant opioid-sparing and co-analgesic properties, dexamethasone should be regarded as a first-line component of such a strategy [36], with negligible side effects associated with a single dose [36,100]. Another primary drug class, widely used and effective, includes 5-HT3 antagonists, applicable for both prophylaxis and rescue therapy [36,101]. Furthermore, emerging and promising evidence suggests that NK-1 receptor antagonists, such as aprepitant, may constitute a potent class of drugs for mitigating PONV [102], with comparative advantages over other antiemetics [91]. These agents are particularly advantageous due to their lack of sedative effects, in contrast to dopamine, histamine, or muscarinic receptor antagonists, which provide a significant benefit in fast-track bariatric surgery within the ERAS protocol [36,93,96,105].

Considering the substantial advancements in recent research regarding various techniques of regional anesthesia, it can be concluded that at least one form of this method, even if limited to local wound infiltration, is recommended for every patient undergoing LBS [15]. Regional anesthesia has been shown to enhance pain management, reduce the requirement for opioids, and consequently diminish opioid-related side effects such as PONV [110]. Nonetheless, discrepancies exist among RCTs evaluating the efficacy of diverse methods, including local wound infiltration, intraperitoneal local anesthetic installation, ESPB, TAP Block, EOI, and QLB, in the prevention of PONV. Often, these studies are designed to compare each technique to a placebo, with limited direct comparisons between specific methods. This represents a significant knowledge gap that necessitates large-scale RCTs or network meta-analyses to facilitate comprehensive comparisons.

In our department, based on our experience, guidelines, and RCTs, we adopted a multimodal approach to optimize PONV prophylaxis and treatment. We implemented a three-drug pharmacological prophylaxis regimen comprising aprepitant 80 mg administered orally one hour prior to surgery, dexamethasone 8 mg given intravenously, and ondansetron 4 mg administered intravenously at the conclusion of the operation, as a rescue therapy. Patients are permitted to consume water up to two hours before the procedure to prevent dehydration. During the intraoperative period, anesthesia is induced and maintained using BIS-guided propofol TIVA and the lowest feasible remifentanil infusion using the target-controlled infusion (TCI) Minto model to maintain hemodynamic stability. We decided against adopting OFA as a standard, considering the limited benefits demonstrated in RCTs conducted within our department [66]. We use TOF-guided rocuronium dosing to achieve neuromuscular blockade with sugammadex reversal as a standard protocol. At the commencement of surgery, surgeons perform a routine local infiltration of the trocar insertion site with 40 mL of 0.25% bupivacaine. Regarding postoperative analgesia, we administer paracetamol and metamizole, and each patient is administered oxycodone 0.1 mg/kg of lean body weight at the end of the surgery. In the PACU, patients may receive additional doses based on the NRS score. We also use the PCA oxycodone administration technique in selected patients.

### 4.1. Limitations of the Study

This scoping review has several limitations. Firstly, there exists significant heterogeneity among the included RCTs, including variation in study design. Overall, no standardized outcome measure has been employed to assess the presence and severity of PONV. Multiple scales are utilized, such as the PONV impact scale or the Likert scale. Furthermore, some studies differentiate between nausea assessment and vomiting, which complicates comparison and hampers definitive conclusions. Additionally, in a minority of RCTs, the Apfel scale is used to predict risk and guide prophylaxis, whereas others do not utilize it. Moreover, our review excluded operational techniques that may influence PONV as well as alternative medicine approaches, which could be components of a multimodal strategy. Aside from these factors, our analysis included exclusively RCTs to mitigate bias; consequently, the scope of the reviewed research, particularly concerning specific techniques, does not encompass all existing published literature. Lastly, only articles published in English were considered, potentially introducing language bias. These methodological limitations warrant cautious interpretation of the findings, which should not be regarded as definitive.

### 4.2. Future Directions

There is limited evidence regarding the optimal drug dosing for co-analgesics such as gabapentinoids or lidocaine, as well as antiemetics, particularly promising NK-1 antagonists. Future research should include large-scale dose-finding RCTs for these agents. Additionally, a substantial knowledge gap exists concerning the efficacy and practical applicability of novel regional blocks such as EOI or QLB in LBS. Finally, given that a broad spectrum of regional anesthesia techniques often cannot be combined due to limitations in the maximum permissible doses of local anesthetics and logistical considerations, a network meta-analysis would be necessary to compare the efficacy of various methods against each other and against a placebo.

## 5. Conclusions

The prevention and management of PONV continue to pose substantial clinical challenges. Although a multimodal strategy is regarded as standard practice for mitigating PONV, debates persist concerning the selection of specific techniques and antiemetic drug regimens. Recent evidence, especially in relation to regional anesthesia techniques and combined pharmacological prophylaxis, including new agents, underlines the potential advantages of innovative approaches.

## Figures and Tables

**Figure 1 jcm-14-06901-f001:**
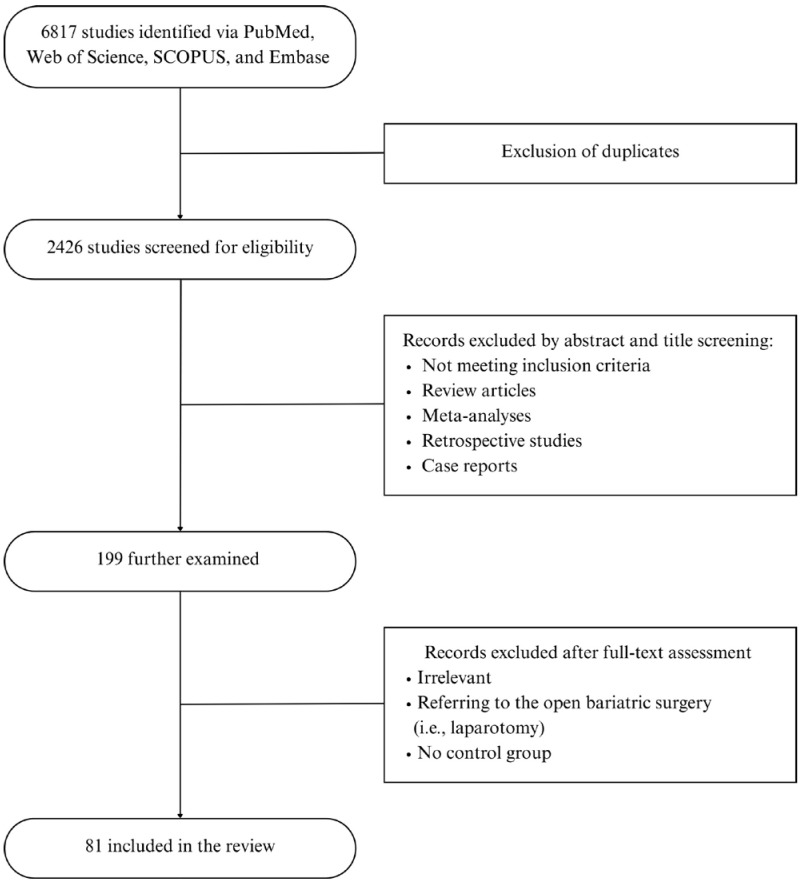
Flow Chart.

**Figure 2 jcm-14-06901-f002:**
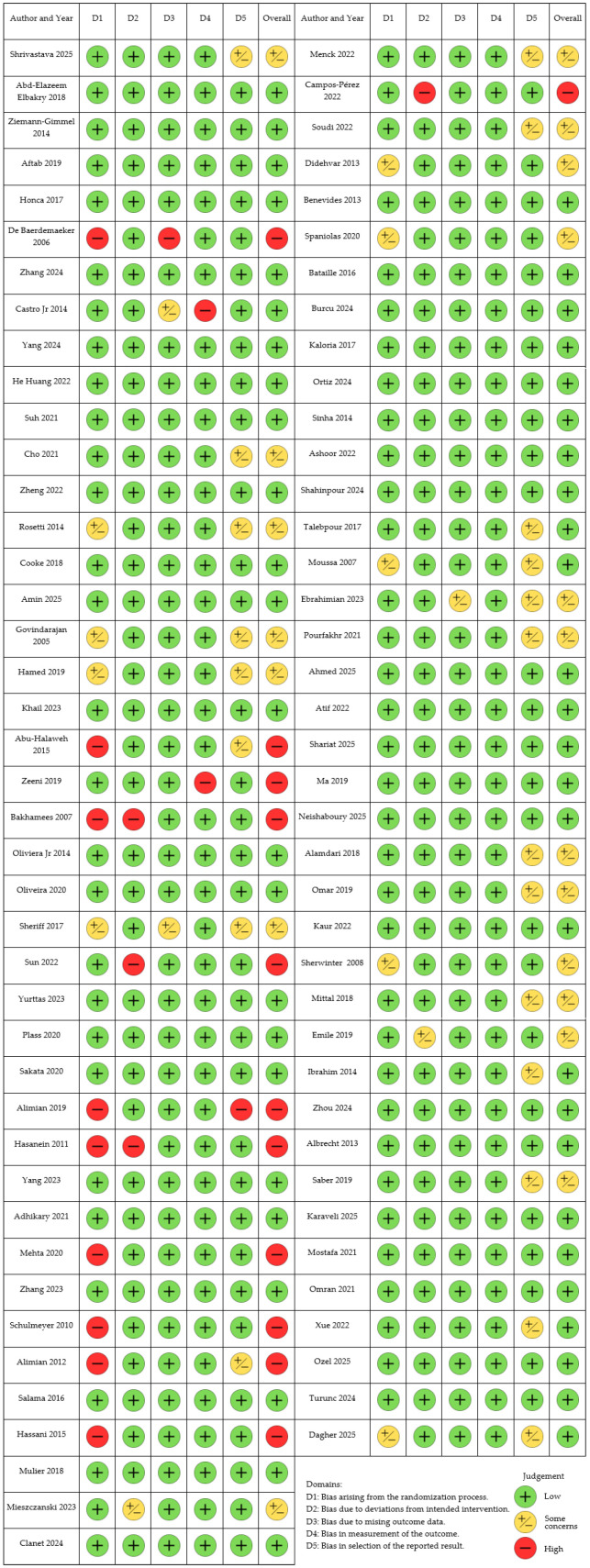
Traffic light plot for Risk of Bias Assessment.

## Data Availability

Data sharing does not apply to this article, as no new datasets were generated or analyzed during the current study. All data analyzed in this review were derived from previously published studies, which are cited within the manuscript. Further information regarding data sources can be found in the references section.

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
