# Peer review of "Evidence-Based Perioperative Prevention of Postoperative Nausea and Vomiting (PONV) in Patients Undergoing Laparoscopic Bariatric Surgery: A Scoping Review"

_jcm, 2025, doi:10.3390/jcm14196901_

Round 1
Reviewer 1 Report
Comments and Suggestions for Authors
Congratulations on this scoping review.
The work is quite comprehensive.
- I think another paragraph should be included (or added to the text) reporting the evidence regarding nasogastric tubes and PONV, starting, for example, from this article : Palomba G, Basile R, Capuano M, Pesce M, Rurgo S, Sarnelli G, De Palma GD, Aprea G. Nasogastric tube after laparoscopic Heller-Dor surgery: Do you really need it? Curr Probl Surg. 2024 Apr;61(4):101457. doi: 10.1016/j.cpsurg.2024.101457. Epub 2024 Feb 15. PMID: 38548426.
- English should be improved
- You should add a short description in the discussion of your experience regarding this matter.
- The complications, duration, and impact of PONV in this surgery should be better described; consider a separate paragraph.
- Another limitation that should be included is whether or not the Apfel score is used.
- The ERAS protocol in the management and minimization of the risk of PONV is often mentioned throughout the paper and should be described clearly and separately.
I look forward to rereading the article with the advice given.
Comments on the Quality of English Language- English should be improved
- Some periods should be summarized.
Author Response
I think another paragraph should be included (or added to the text) reporting the evidence regarding nasogastric tubes and PONV, starting, for example, from this article : Palomba G, Basile R, Capuano M, Pesce M, Rurgo S, Sarnelli G, De Palma GD, Aprea G. Nasogastric tube after laparoscopic Heller-Dor surgery: Do you really need it? Curr Probl Surg. 2024 Apr;61(4):101457. doi: 10.1016/j.cpsurg.2024.101457. Epub 2024 Feb 15. PMID: 38548426.
Our response: Thank you for your suggestion. We have included a dedicated paragraph in the text. The issue of nasogastric tube placement is a crucial clinical concern and should not be omitted.
English should be improved
Our response: To improve our English and provide the highest quality, we commissioned language editing and performed careful proofreading. We hope that it is now optimal.
You should add a short description in the discussion of your experience regarding this matter.
Our response: We agree with your suggestion, and an additional description has been added at the end of the discussion, sharing our own experience in that field.
The complications, duration, and impact of PONV in this surgery should be better described; consider a separate paragraph.
Our response: We have expanded the part describing complications, duration, and impact of PONV into a separate paragraph.
Another limitation that should be included is whether or not the Apfel score is used.
Our response: We have included it among the limitations, adhering to your suggestion.
The ERAS protocol in the management and minimization of the risk of PONV is often mentioned throughout the paper and should be described clearly and separately.
Our response: In compliance with your advice, we have described ERAS recommendations, including PONV prophylaxis, in a separate paragraph.
I look forward to rereading the article with the advice given.
Our response: Thank you for your comments. We tried to implement them to improve the quality of our manuscript. 
Reviewer 2 Report
Comments and Suggestions for Authors
I would like to thank the authors for sharing this scoping review on the prevention of postoperative nausea and vomiting in patients undergoing laparoscopic bariatric surgery. This is a very relevant topic, since PONV is both common and problematic in this setting, with a clear impact on recovery and postoperative outcomes. The idea of systematically collecting and organizing the available evidence is therefore very valuable.
I particularly appreciated that the literature search was broad and that the study selection process was clearly described and transparent. A large number of RCTs were included, and the results were organized in a way that makes it easier to understand the different strategies, from anesthetic choices to antiemetic drugs and regional techniques. It is also useful that the review is framed in relation to ERABS recommendations and highlights the importance of a multimodal approach.
That said there are some aspects that would benefit from clarification. The decision to include only RCTs in English should be explained more thoroughly. It is understandable to focus on high quality studies, but at the same time this restriction may exclude relevant evidence and introduce bias. Another point concerns the definition of outcomes: the included trials used different criteria and time frames for assessing nausea and vomiting, which makes direct comparisons difficult. A summary table showing, for each trial, how PONV was defined and at what time points would be very helpful.
I also found it very interesting that the review points out how often interventions are combined, making it difficult to attribute the effect to a single component. This issue affects many of the included studies, and I think it could be emphasized even more, perhaps with a short paragraph in each section addressing the risk of confounding by cointerventions.
Finally adding even a qualitative assessment of the methodological quality of the included RCTs would strengthen the conclusions. While not mandatory for a scoping review, in this context it would add real value
There are a few details that should be corrected: for example, in one place the name of an author appears as “Zimman-Gimmel” instead of “Ziemann-Gimmel,” and in one of the tables “Ondanestron” is written instead of “Ondansetron.” These are small errors but worth correcting to make the text cleaner.
In conclusion, I found this review useful and interesting, with clear take-home messages and practical relevance. Before publication, I would recommend addressing the points above to strengthen the methodology and improve readability. After these revisions, the manuscript could become an important reference for clinicians managing PONV in bariatric surgery.
Author Response
That said there are some aspects that would benefit from clarification. The decision to include only RCTs in English should be explained more thoroughly. It is understandable to focus on high quality studies, but at the same time this restriction may exclude relevant evidence and introduce bias.
Our response: Thank you for your suggestion. In accordance with it, we have provided a more detailed explanation for including only RCTs in English.
Another point concerns the definition of outcomes: the included trials used different criteria and time frames for assessing nausea and vomiting, which makes direct comparisons difficult. A summary table showing, for each trial, how PONV was defined and at what time points would be very helpful.
Our response: We agree that incorporating an outcome measure indicating the method and timing of PONV assessment is essential. Consequently, we have added a column to each table with such information.
I also found it very interesting that the review points out how often interventions are combined, making it difficult to attribute the effect to a single component. This issue affects many of the included studies, and I think it could be emphasized even more, perhaps with a short paragraph in each section addressing the risk of confounding by cointerventions.
Our response: We agree on that point. Therefore, we added a separate paragraph addressing that issue in the Results section. We also aimed to provide a more thorough description of the interventions in the study group in the tables.
Finally adding even a qualitative assessment of the methodological quality of the included RCTs would strengthen the conclusions. While not mandatory for a scoping review, in this context it would add real value
Our response: Adhering to your suggestion, we performed and included methodological quality assessment using the Cochrane Risk of Bias Tool. We added a paragraph and the RCTs assessment in Figure 2.
There are a few details that should be corrected: for example, in one place the name of an author appears as “Zimman-Gimmel” instead of “Ziemann-Gimmel,” and in one of the tables “Ondanestron” is written instead of “Ondansetron.” These are small errors but worth correcting to make the text cleaner.
Our response: We conducted thorough proofreading and commissioned a language edition to enhance readability and eliminate errors.
In conclusion, I found this review useful and interesting, with clear take-home messages and practical relevance. Before publication, I would recommend addressing the points above to strengthen the methodology and improve readability. After these revisions, the manuscript could become an important reference for clinicians managing PONV in bariatric surgery.
Our response: Thank you for your comments. We tried to implement them to improve the quality of our manuscript. 
Round 2
Reviewer 1 Report
Comments and Suggestions for Authors
I was pleased to read your changes and suggestions.
Congratulations again on your work.
Reviewer 2 Report
Comments and Suggestions for Authors
I am satisfied with the revisions performed by the authors. The manuscript has been substantially improved.